

# Proanthocyanidins in seed coat tegmen and endospermic cap inhibit seed germination in *Sapium sebiferum*

Faheem Afzal Shah[1],*, Jun Ni[2],*, Jing Chen[2], Qiaojian Wang[1], Wenbo Liu[2], Xue Chen[2], Caiguo Tang[2], Songling Fu[1] and Lifang Wu[2]

[1] School of Forestry and Landscape Architecture, Anhui Agriculture University, Hefei, Anhui, China
[2] Key Laboratory of High Magnetic Field and Ion Beam Physical Biology, Hefei Institutes of Physical Science, Chinese Academy of Sciences, Hefei, Anhui, China
* These authors contributed equally to this work.

Corresponding authors
Songling Fu,
fusongl001@outlook.com
Lifang Wu, lfwu@ipp.ac.cn

## ABSTRACT

*Sapium sebiferum*, an ornamental and bio-energetic plant, is propagated by seed. Its seed coat contains germination inhibitors and takes a long time to stratify for germination. In this study, we discovered that the *S. sebiferum* seed coat (especially the tegmen) and endospermic cap (ESC) contained high levels of proanthocyanidins (PAs). Seed coat and ESC removal induced seed germination, whereas exogenous application with seed coat extract (SCE) or PAs significantly inhibited this process, suggesting that PAs in the seed coat played a major role in regulating seed germination in *S. sebiferum*. We further investigated how SCE affected the expression of the seed-germination-related genes. The results showed that treatment with SCE upregulated the transcription level of the dormancy-related gene, gibberellins (GAs) suppressing genes, abscisic acid (ABA) biosynthesis and signalling genes. SCE decreased the transcript levels of ABA catabolic genes, GAs biosynthesis genes, reactive oxygen species genes and nitrates-signalling genes. Exogenous application of nordihydroguaiaretic acid, gibberellic acid, hydrogen peroxide and potassium nitrate recovered seed germination in seed-coat-extract supplemented medium. In this study, we highlighted the role of PAs, and their interactions with the other germination regulators, in the regulation of seed dormancy in *S. sebiferum*.

## INTRODUCTION

Seed germination is an important step in the plant life-cycle because it determines subsequent plant survival and reproductive success. The seed coat can play a role in regulating dormancy. Known mechanisms by which the seed coat regulates dormancy include the prevention of water uptake by an impermeable seed coat, or inhibiting gas exchange (*McGill et al., 2017*), or by mechanically constraining the embryo, or by containing germination inhibitors (*Baskin & Baskin, 2014*). Proanthocyanidins (PAs) are the chemical inhibitors of germination found in the seed coat of many plants. As early as *1914*, Nilsson-Ehle showed that red seed coat colour in wheat is associated with extended dormancy period in comparison to that of white-grained wheat. In *1958*,

Miyamoto & Everson showed that the red pigment of the seed coat is a substance derived from catechins, which make up PAs. Red seeds of charlock (*Sinapis arvensis L.*) exhibit a reduced dormancy compared with black seeds (*Durán & Retamal, 1989*). In legumes, coloured seeds imbibe slower than white seeds and then germinate later (*Kantar, Pilbeam & Hebblethwaite, 1996*; *Powell, 1989*; *Werker, Marbach & Mayer, 1979*; *Wyatt, 1977*). In *Rubus* seed, PAs contribute to seed coat hardness and resulting seed dormancy (*Wada, Kennedy & Reed, 2011*). In *Arabidopsis*, the permeability and thickness of the testa are affected by the PAs and some structural elements altered in mutants, which may lead to effects on germination (*Debeaujon, Leon-Kloosterziel & Koornneef, 2000*). The seeds need gibberellins (GAs) to overcome the dormancy imposed by the seed coat (*Debeaujon & Koornneef, 2000*). PAs located in the seed coat can act as a doorkeeper to seed germination. Inhibitory effect of PAs on seed germination is due to de novo biogenesis of abscisic acid (ABA). Compared with wild-types, PA-deficient mutants contain a lower concentration of ABA during germination. PA regulation of seed germination is mediated by the ABA signalling pathway (*Himi et al., 2002*; *Liguo et al., 2012*). PAs modulate the activities of the Class III peroxidase that controlled the levels of reactive oxygen species (ROS) during seed germination (*Jia et al., 2012*, *2013*).

Chinese tallow (*Sapium sebiferum* L.) belongs to the Euphorbiaceae family and is native to eastern Asia (*Esser, 2002*). It is popular because of its colourful autumn foliage (*Zhao & Tao, 2015*). The white waxy aril of the seeds contains highly saturated fatty acids and highly unsaturated oil is found in the seed (*Boldor et al., 2010*). Tallow has been used for manufacturing soap, candles, cloth and fuel, while the seed's oil can be used for making native paints and varnishes (*Brooks et al., 1987*; *Jeffrey & Padley, 1991*). A single mature tree of *S. sebiferum* produces many seeds. The estimated yield of a *S. sebiferum* tree is 4,700 L of oil per hectare every year which far exceeds the average commercial yields of traditional oilseed crops (*Boldor et al., 2010*; *Webster, Jenkins & Jose, 2006*). That is why, *S. sebiferum* has recently become a species of interest as a source of biodiesel (*Gao et al., 2016*).

Sexual propagation is an easy method of commercial propagation, and it's being used widely for the commercial propagation of a large number of plant species, including many bio-energetic plants like *S. sebiferum*. However, the poor rate of seed germination due to deep dormancy has seriously limited its use (*Li et al., 2012*).

*Sapium sebiferum* seeds have hard, dark brown to blackish seed outer testa and reddish brown inner tegmen. The tegmen encloses the endosperm, which in turn encloses the embryo. Tallow tree seeds readily imbibed water but the seed coat at the site of the radicle appeared to be a barrier to seed germination. Germination of cabbage seeds was inhibited when cabbage seeds were soaked in extracted solutions from *S. sebiferum* seed coat (*Li et al., 2012*). Moreover, it has also been discovered that endosperm extracts have a stronger inhibitory effect on cabbage seed germination than seed coat extracts (SCEs) of *S. sebiferum* (*Qian et al., 2016*). We hypothesize that germination inhibitors found in *S. sebiferum* could be the PAs. It is currently unknown which layer of the seed coat has the highest concentration of PAs. PAs inhibit seed germination by influencing ABA, GA and ROS regulatory genes (*Debeaujon & Koornneef, 2000*; *Debeaujon, Leon-Kloosterziel & Koornneef, 2000*; *Jia et al., 2012*, *2013*; *Liguo et al., 2012*). Nitrates also play an important

role during seed germination (*Lara et al., 2014*). It is unclear whether PAs respond to nitrate signalling. We tested whether exogenous application of an ABA biosynthesis inhibitor nordihydroguaiaretic acid (NDGA), gibberellic acid (GA$_3$), hydrogen peroxide (H$_2$O$_2$) and potassium nitrate (KNO$_3$) promote seed germination is the presence of SCE (which may contain PAs inside). We conducted several experiments to address these questions, to test our hypothesis and to demonstrate the mechanism involved in *S. sebiferum* seed dormancy.

## MATERIALS AND METHODS

### Seed material collection and storage

*S. sebiferum* seeds were harvested from 6 plants grown in the experimental field of Hefei Institute of Physical Science, Chinese Academy of Sciences (31°52°02N, 117y17 07E), Anhui, China. Hefei has a humid subtropical climate with four distinct seasons. According to Anhui's meteorological bureau, Hefei's annual average temperature is 16.2 °C. Hefei's annual average low temperature is 12.6 °C. Summers are hot and humid, with a July average of 28.3 °C. Its annual precipitation is over 1,000 mm. Seeds were collected in December 2016 from 6 trees, filled in nylon bags and stored at room temperature prior to use. The experiment was conducted from March to October 2017.

### Chemicals and stock solutions

The GA$_3$ (CAS# 77-06-5), NDGA (≥97%, CAS# 500-38-9) and Vanillin (CAS# 121-33-5) were purchased from Sigma-Aldrich (Shanghai) Trading Co., Ltd., Shanghai, China. KNO$_3$ (CAS# 7757-79-1), H$_2$O$_2$ (30%, CAS# 7722-84-1), sulphuric acid (H$_2$SO$_4$, CAS 7664-93-9) and hydrochloric acid (HCl, CAS# 7647-01-0) were purchased from Sinopharm Chemical Reagent Co., Ltd., Shanghai, China. Sodium hypochlorite (NaClO, CAS# 7681-52-9) was bought from Sangon Biotech (Shanghai) Co., Ltd., Shanghai, China. A pure supply of PAs (UV ≥ 95%, CAS 4852-22-6) was purchased from Shanghai Aladdin Biochemical Technology Co., Ltd., Shanghai, China. Ethanol (99.7%, CAS# 64-17-5) and methanol (99.7%, CAS# 67-56-1) was purchased from Shanghai Titan Scientific Co., Ltd., Shanghai, China. Sodium hydroxide (NaOH, CAS# 1310-73-2) was purchased from Shanghai Chemical Reagent Co., Ltd., Shanghai, China. Murashige and Skoog (MS) medium was purchased from Qingdao Hope Bio-Technology Co., Ltd., Shandong, China. The GA$_3$ and NDGA stock solutions (100 mM) were prepared by dissolving in 80% methanol. KNO$_3$ stock solution (10%) was prepared by dissolving in distilled water. All stock solutions were diluted in distilled water to make working solutions.

### Seed coat extraction, application and PAs analysis

Seed coat extract was prepared as described by *Li et al. (2012)* with little modification. *S. sebiferum* tree seed coats were ground into powder. Ten grams of seed coat powder was dissolved in 200 mL of 80% (v/v) aqueous methanol and placed in a refrigerator at 4.0 °C for 24 h. After centrifugation (Allegra X-30R Centrifuge; Beckman Coulter, Inc. 250 S. Kraemer Boulevard Brea, CA, USA) at 4,500 rpm at 4.0 °C for 10 min, the supernatant was evaporated under vacuum at 40.0 °C.

Seed coat extract and PA medium for growing seeds were prepared by using 0.1%, 0.2% and 0.3% SCE and pure PAs in 0.5× Morashige and Skoog medium (MS) containing 15 g/L sucrose and 8 g/L agar before sterilization. All media were autoclaved at 121.0 °C for 22 min and poured into 9 cm diameter Petri dishes (20 mL each) under a laminar flow hood. Seed coat proanthocyanidin contents were analysed by conventional HCl–vanillin assay (*Herald et al., 2014*). Seed coat was ground under liquid nitrogen by a YLK Ball Mill Machine (YLT-04, Hunan, China), and 30 mg seed-coat powder was extracted for 20 min in 5 mL 1% HCl in methanol at 30.0 °C in a water bath. The extracts were centrifuged at 4,500 rpm for 5 min. One mL aliquots of the extract were dispensed into 2 culture tubes designated as sample and sample control. The tubes were incubated in a water bath at 30.0 °C for approximately 5 min. A working vanillin reagent was prepared daily by mixing equal amounts of 1% vanillin with 8% HCl solutions. Five mLs of the working vanillin reagent were added at 1 min intervals to the extract, and 5.0 mLs of 4% HCl were added to the sample control. The prepared tubes were incubated in a water bath at 30.0 °C for exactly 20 min, after which the absorbance was measured at 500 nm using nano-spectrophotometer of ScanDrop 100 (Analytik Jena AG, Überlingen, Germany). Final absorbance was calculated by subtracting the absorbance of the sample control from the corresponding vanillin-containing sample. Standard curves were developed using pure PAs (UV ≥ 95%) at concentrations that ranged from 0 to 100 µg/mL. ESC's PAs were analysed by the previously used protocol as described by *Xuan, Wang & Jiang (2014)*. Decoated seeds were dipped in 1% vanillin with 8% HCl solution and incubated in a water bath at 30.0 °C for 20 min. Photographs were taken by an Olympus SZX10 stereomicroscope having a TUCSEN 6.0 megapixel USB 2.0 colour camera.

## Pre-germination treatments and germination conditions

Seeds were washed with 1% sodium hydroxide to remove white tallow. Sulphuric acid scarification was done by dipping seeds in 98.08% concentrated sulphuric acid at 4.0 °C for 10-, 20-, 30-, 40-, 50- and 60 min. After each time period of sulphuric acid treatment, seeds were washed in running tap water 5 times. For mechanical scarification, a cut was made by scissors in the seed coat on the opposite side of the radical. Intact seeds and scarified seeds were sown separately in 5 replicates (10 seeds in each pot) in 10 × 10 cm pots containing peat moss. We investigated the water uptake of intact seed and sulphuric acid scarified seed using the method of *Li et al. (2012)*.

The seed coat was removed by dissection leaving the embryo (embryonic axis and endosperm) as a single unit, hereafter referred to as a 'decoated seed.' We used decoated seed to verify the effects of PAs and SCEs. Decoated seeds were sterilized by washing twice with 70% ethyl alcohol for 30 s and then incubated in 20% NaClO for 10 min. After rinsing off NaClO, the seeds were washed 3 times with autoclaved water and dried by blotting over sterilized filter papers. For the ESC experiments, ESC was removed with a sterilized blade in a laminar flow hood.

Sterilized decoated seeds were sown in SCE and PA mediums. To investigate the relationship of seed coat PAs with GA-, ABA-, ROS- and nitrates-signalling genes, we primed the decoated seeds in sterile water (for control), GA$_3$ (50 µM), NDGA (50 µM),
$H_2O_2$ (20 mM) and 0.4% $KNO_3$ separately overnight (12 h) at room temperature (25.0 °C) and sowed the primed seed in 0.5× MS medium supplemented with 0.3% SCE. Germination conditions for all experiments were maintained as day/night temperatures of 25.0/20.0 °C, with 16 h light/8 h dark photoperiod, 150 μmol/m²/s photosynthetic photon flux density and 70% relative humidity. Protrusion of the radical from the micropyle was considered as the standard for seed germination. Germination data were recorded every day after germination started (5 days after imbibition). Shoot and root length were measured manually with a ruler. All seed germination pictures were taken by NIKON D90 containing NIKON DX AF-S NIKKOR 18-105 mm lens (NIKON Corporation, Tokyo, Japan).

### Primer designing, RNA extraction, cDNA synthesis and RT–qPCR conditions

The full sequences of all genes were obtained by local blasting Arabidopsis amino-acids sequence in blast-2.2.31. A local blast library was built by flower-bud transcriptome (Accession: SRX656554, https://www.ncbi.nlm.nih.gov/sra/SRX656554) of *S. sebiferum* (*Yang et al., 2015*). The list of all genes' full mRNA sequences is available in Data S1. Primers used for qPCR were designed by using primer premier 6. The $T_m$ of the primers was between 59.0 and 61.0 °C and a list of all primers are given in Table S1. For gene expression analysis, seed samples were taken the third and sixth day after imbibition. Samples were frozen in liquid nitrogen and stored at −80.0 °C. RNA was extracted by using E.Z.N.A® plant RNA extraction kit (Omega Bio-tek, Inc., Norcross, GA) according to the given protocol. Five hundred nanograms RNA of each sample was reverse transcribed using cDNA synthesis SuperMix (TransGen Biotech., Shanghai, China) according to the given protocol. The cDNA samples were diluted 25× with sterile water. For each qPCR, 9 μL of the sample, 10 μL of the 2× QuantiNova SYBR Green PCR Master Mix (QIAGEN, Pudong, Shanghai, China) and 0.5 μL of each primer was added to make a final volume of 20 μL. The RT–qPCRs were run on a Light Cycler®96 (Roche Diagnostics, Indiana, USA). The qPCR program run consisted of the first step at 95.0 °C for 3 min and afterwards 45 cycles alternating between 15 s at 95.0 °C, 15 s at 60.0 °C and 15 s at 72.0 °C.

### Statistical analysis

All data was arranged in Excel 2013 and statistical analyses were done in R Studio 1.1.383. All data were represented with mean ± standard deviation. Results from the different treatments were analysed separately. The significance of treatments was tested by one-way analysis of variance (ANOVA). Duncan's multiple range test and the Tukey test were used to identify significant differences between pairs of means at $P < 0.05$.

## RESULTS

### Sulphuric acid scarification promotes seed germination

We found that sulphuric acid digested the seed coat external surface and caused cracks. About 10 and 20 min incubation times digested the epidermal layer of the seed coat while the 30-, 40- and 50 min incubations caused mild cracks in the seed coat. But the 60 min incubation in sulphuric acid caused deep cracks in the seed coat (Fig. 1A).

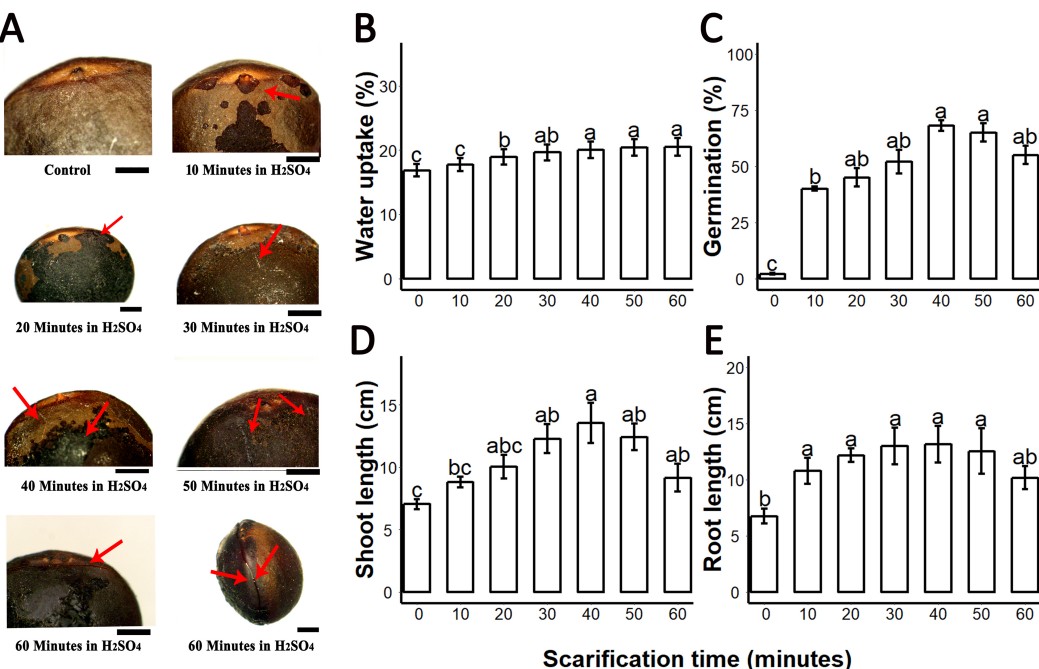

**Figure 1 Sulphuric acid (SA) scarification significantly promoted the seed germination of *Sapium sebiferum.*** (A), Effect of SA scarification time on seed coat; red arrows indicate the bruises, scars and cracks caused by SA. Bars 1 mm. (B), SA impacts on water uptake in the seed. (C), SA-induced seed germination of *S. Sebiferum.* (D and E), Impact of SA on shoot and root length of seedlings respectively. Shoot and root length were measured after 45 days of imbibition. Data shown are means ± SD (*n* = 3). Means with different letters are significantly different at *P* < 0.05 using Duncan's multiple range HSD post hoc test. The photographs were taken by Shah Faheem Afzal and Jun Ni.

When we measured the PA contents of 0-, 10-, 20-, 30-, 40-, 50- and 60 min scarified seed, we found that sulphuric acid scarification degradation of the PA contents proportionally to the incubation time (Fig. 2).

We investigated the water uptake of intact seeds and sulphuric-acid scarified seeds. Water uptake of untreated control seeds was 16.5–17.8% over 72 h; seeds treated with sulphuric acid for longer periods of time showed increasing water uptake to a maximum of 22.5% after 60 min of acid treatment. Our results showed that water uptake gradually increased from untreated (control) 0–10 min, 20-, 30-, 40-, 50- and 60 min. Water uptake percentages of 20-, 30-, 40-, 50- and 60 min scarified seed were significantly different from the control, but the water uptake percentage of 10 min scarified was not significantly different from control (Fig. 1B). We found that germination of intact seeds was 2%, and ranged from 40 to 65% in sulphuric-acid-scarified seeds while mechanically scarified showed 20% germination in 20 days. (Fig. 1C; Fig. S1). Intact and scarified seed's germination were significantly different (*P* = 0.05).When we measured the root and shoot length of seedling of 45-day-old seedlings, we found that the seedlings whose seeds were chemically scarified by 30-, 40- and 50 min incubation in sulphuric acid, had more root and shoot length than the seedlings of 0-, 10-, 20- and 60 min incubated seeds in sulphuric acid (Figs. 1C and 1D).

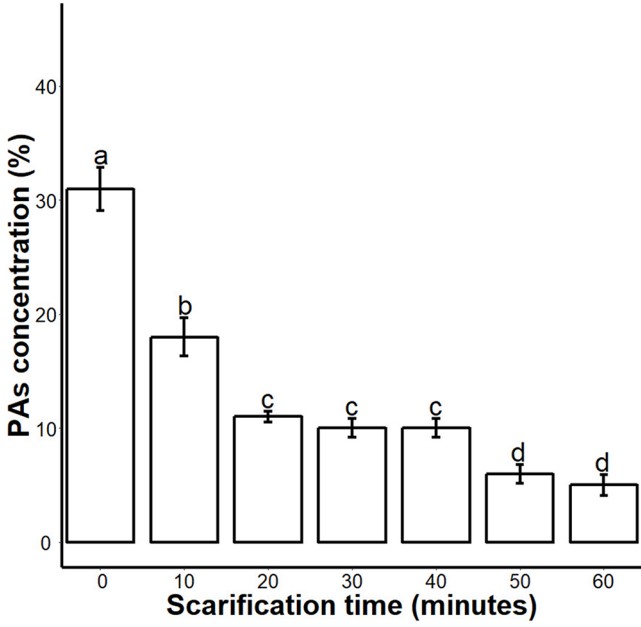

**Figure 2 Impact of sulphuric acid scarification on PA contents of *S. Sebiferum* seed coat.** Seeds of *S. Sebiferum* were dipped in concentrated sulphuric acid for 10, 20, 30, 40, 50 and 60 min separately. PA contents of acid-scarified seeds were determined by vanillin assay. Data shown are means ± SD ($n = 3$). Means with different letters are significantly different at $P < 0.05$ using Tukey's HSD post hoc test.

## Tegmen and ESC contained PAs which inhibited seed germination

From germination analysis, we found that the decoated seeds showed 85 ± 5% seed germination within seven days (Fig. 3B; Fig. S2). SCE supplemented medium inhibited seed germination (Fig. 3B). We determined PA and SCE concentration in *S. sebiferum* seed coat testa and tegmen separately. We found that testa and tegmen contain 3 ± 2% and 65 ± 5% (mean ± SD) of PAs respectively, while SCE contained 30 ± 3% (mean ± SD) PAs (Fig. 3A). We also tested seed germination in proanthocyanidins-supplemented 0.5× MS, the results showed that PAs significantly inhibited the seed germination in *S. sebiferum* (Fig. 3B).

Further, we found that the seeds of *S. sebiferum* contain a dark brownish colour ESC. When cultivated on 0.5× MS, we found that dark brownish ESC became darker in some dormant seeds (Fig. 4A). When we removed that ESC and cultivated those ESC removed seed on 0.5× MS, we found the seed without ESC showed 100% seed germination within 5 days which was significantly different than the seed with ESC (Figs. 4B and 4C). We hypothesized that the dark brownish ESC might accumulate PAs, which inhibited seed germination. Then we determined PAs by vanillin assay and interestingly we found that ESC gives red colour which is an indication of PAs. We also tested the dynamic changes in PAs in ESC of intact seeds cultivated in peat moss media. We found that the intensity of PAs gradually decreased with imbibition time, and, after complete diminishing of PAs in ESC, the seed showed signs of germination (Fig. 4D).

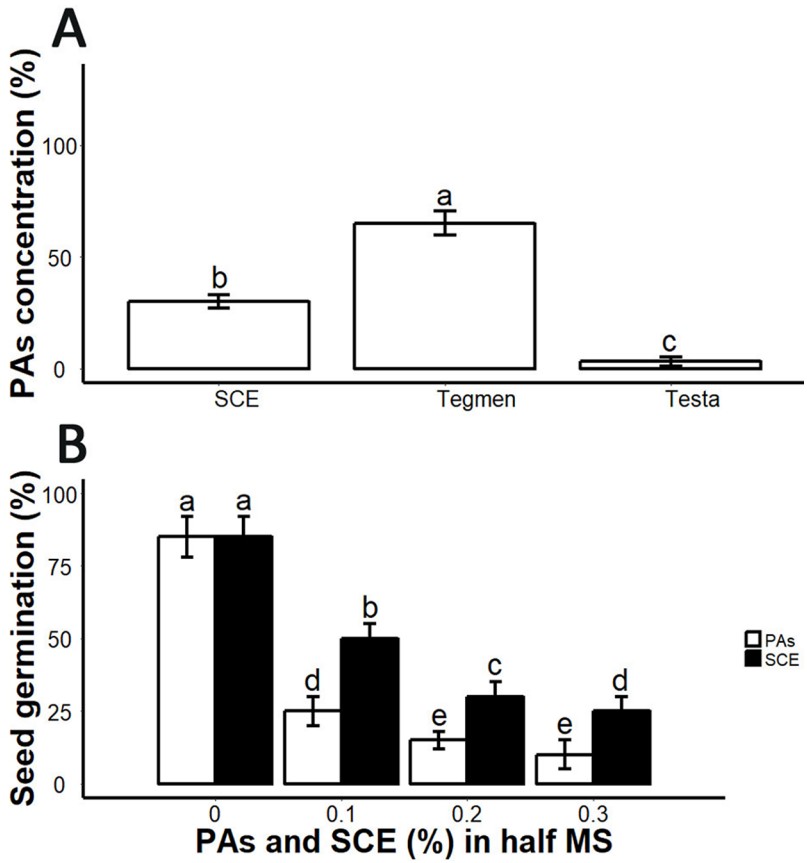

**Figure 3 Impacts of exogenous application of SCE and PA on seed germination.** (A), PA contents in SCE, tegmen and testa of *S. Sebiferum* seed coat. (B), Impact of different concentrations of SCE and PAs on seed germination. Data shown are means ± SD ($n = 3$). Means with different letters are significantly different at $P < 0.05$ using Tukey's HSD post hoc test.  

## Effect of SCE on the expression level of dormancy-related genes

We examined the relationship between the seed coat dormancy and the expression levels of GA-, ABA-, ROS- and nitrates-related genes. To investigate whether this effect was because of PAs, we compared the relative expression of GA-, ABA-, ROS-, nitrates- and dormancy-related genes between the control, SCE and PA treatments. It is very important to check the expression of dormancy-specific genes while studying seed dormancy. *Delay of Germination 1 (DOG1)* is a dormancy-specific gene which positively regulates seed dormancy (*Dekkers et al., 2016*; *Footitt et al., 2017*). We found that the expression level of *SsDOG1* was significantly higher in SCE and PAs as compared to control on the third and sixth days of imbibition. But the expression level of *SsDOG1* at both time points was not significantly different between SCE and PAs (Fig. 5A).

It has been reported that dormant seeds have high levels of ABA (*Millar et al., 2010*). To find the transcriptional changes of ABA-related genes during different imbibition periods of different treatments, we selected 9-cis-epoxycarotenoid dioxygenases 6 (*NCED6*), *INSENSITIVE3* (*ABI3*) and *CYP707A2* as ABA biosynthesis, signalling and catalyzing genes, respectively (*Dekkers et al., 2016*; *Footitt et al., 2011*). We also found that PAs and

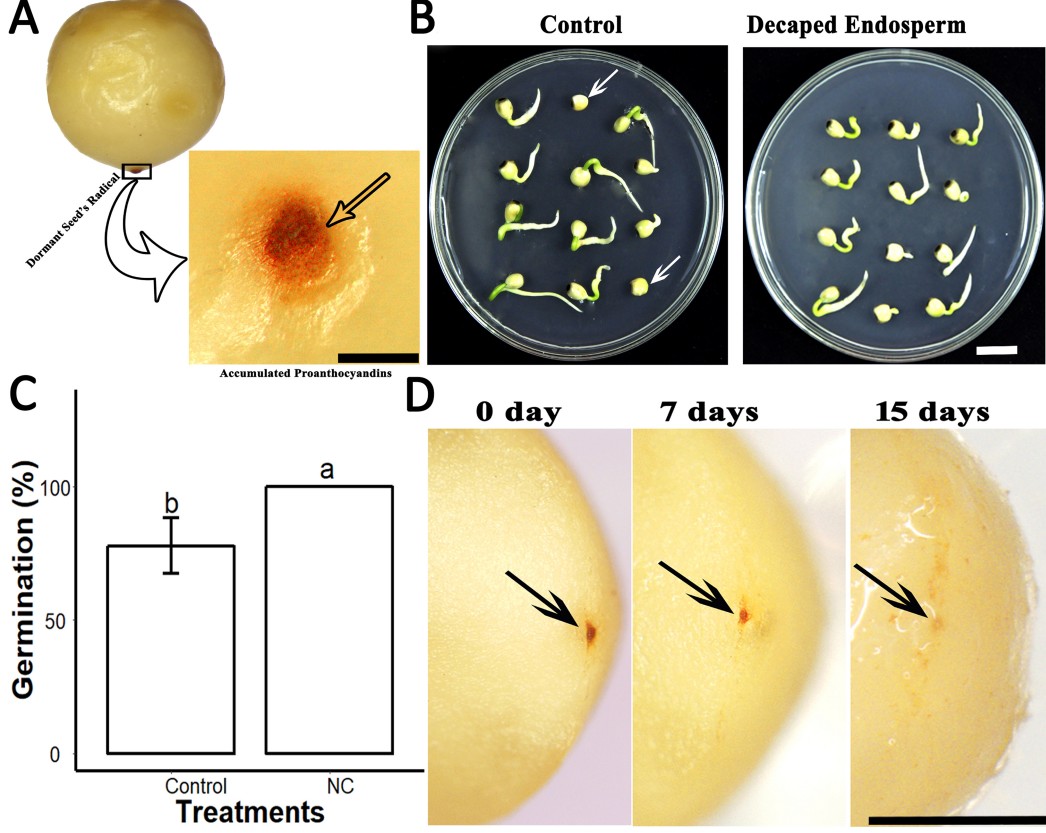

**Figure 4 PAs in the endospermic cap affected the seed germination.** (A), Accumulation of PAs in endospermic cap of dormant seed. (B and C), Decaping of endospermic cap significantly promoted seed germination as compared to control (with endospermic cap). (D), Dynamic changes of PAs in the endospermic cap of non-dormant seed. Bars in (A) and (D) 2 mm, (B) 1 cm. The photographs were taken by Shah Faheem Afzal.

SCE both promoted the expression level of *SsNCED6* on both the third and sixth days of imbibition (Fig. 5B). In SCE and PAs, the *SsCYP 707A2* expression decreases gradually with time, while in the control *SsCYP 707A2* expression is higher during both third and sixth days of imbibition (Fig. 5C). Expression levels of *SsABI3* were not significantly different between control, PAs and SCE on the third day of imbibition. Interestingly, on the sixth day of imbibition, the *SsABI3* expression level remained the same in SCE and PAs but dropped in the control (Fig. 5D).

Gibberellins are plant hormones that play an important role in seed germination. we selected a GA-biosynthesis gene (*GA3OX1*), a GA-inactivating gene (*GA2OX*), GA negative regulator genes *GAI* (*GIBBERELLIC ACID INSENSITIVE*) and *RGL2* (*REPRESSOR-OF-GA1 2*) (*Lee et al., 2002*; *Matsushita et al., 2007*; *Ravindran et al., 2017*; *Rieu et al., 2008*; *Shen et al., 2016*). PAs and SCE significantly repressed the *SsGA3OX1* expression level as compared to the control on the third day, while *SsGA3OX1* transcript remained unchanged on the sixth day in both SCE and PAs. Expression levels of *SsABI3* were not significantly different between control, PAs and SCE on the third day of imbibition. Interestingly, on the sixth day of imbibition, the *SsABI3* expression level

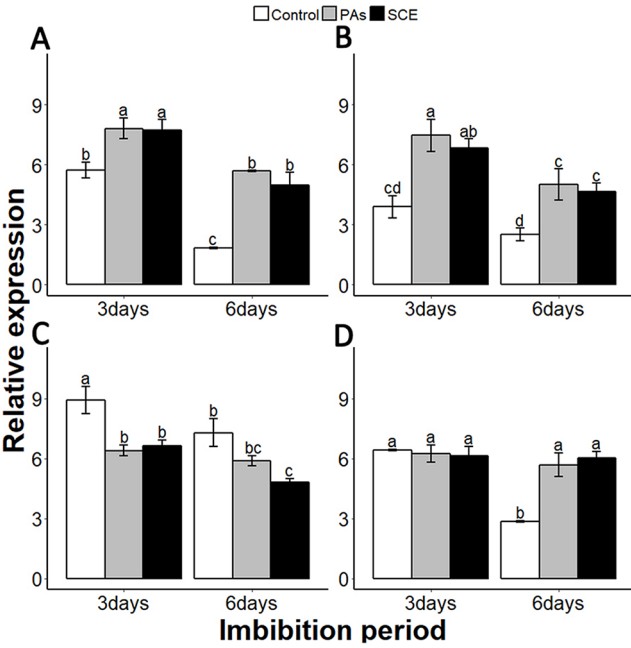

**Figure 5 Effect of SCE on the expression of seed dormancy-related gene (*SsDOG1*) and ABA-related genes.** (A), *SsDOG1* (B), *SsNCED6* (C), *SsCYP707A2* and (D), *SsABI3*. The expression of *SsDOG1*, *SsNCED6*, *SsCYP707A2* and *SsABI3* were determined by qRT-PCR on third and sixth day after treatment. *SsACTIN* was used as the reference gene. Control, seed grown in half MS medium. PAs, proanthocyanidins-supplemented half MS medium. SCE, half MS medium supplemented with seed-coat extract. Data shown are means ± SD ($n = 3$). Means with different letters are significantly different at $P < 0.05$ using Tukey's HSD post hoc test.

remained the same in SCE and PAs but dropped in the control (Fig. 5D). In PAs and SCE, the expression level of *SsGA2OX* increased on the third day and then decreased insignificantly on the sixth day of imbibition (Fig. 6A). In the control treatment, *SsGA2OX* transcription decreased gradually with time (Fig. 6C). *SsRGL2* and *SsGAI* expression levels were higher in SCE and PAs as compared to the control during both the third and the sixth days of imbibition (Figs. 6B and 6D, respectively).

Reactive oxygen species are highly active during seed germination. *MITOGEN-ACTIVATED PROTEIN KINASE* (*MPKs*) protein regulates the ROS signalling. Among the *MPKs* genes, *MPK6* is highly active during seed germination (*Oracz et al., 2009*; *Oracz & Karpinski, 2016*). In our experiments, the effects of PAs and SCE were negative on *SsMPK6* transcription. Relative expression of the *SsMPK6* decreased over time in the seed growing on SCE and PAs. On the other hand, in the control treatment, the transcription levels of the *SsMPK6* increased from the third to the sixth days of imbibition (Fig. 7A).

Among the soil nutrients, nitrates play an important role in seed germination with a specific molecular mechanism (*Lara et al., 2014*). To investigate the impact of seed coat on nitrates signalling, we selected nitrates-signalling genes *NIN-LIKE-PROTEIN 8* (*NLP8*) and *CBL-INTERACTING PROTEIN KINASE 23* (*CIPK23*) (*Footitt et al., 2017*; *Yan et al., 2016*). The transcript level of *SsNLP8* was higher in the control as compared to PAs and SCE treatments in both third and the sixth days of imbibition (Fig. 7B). PAs and SCE

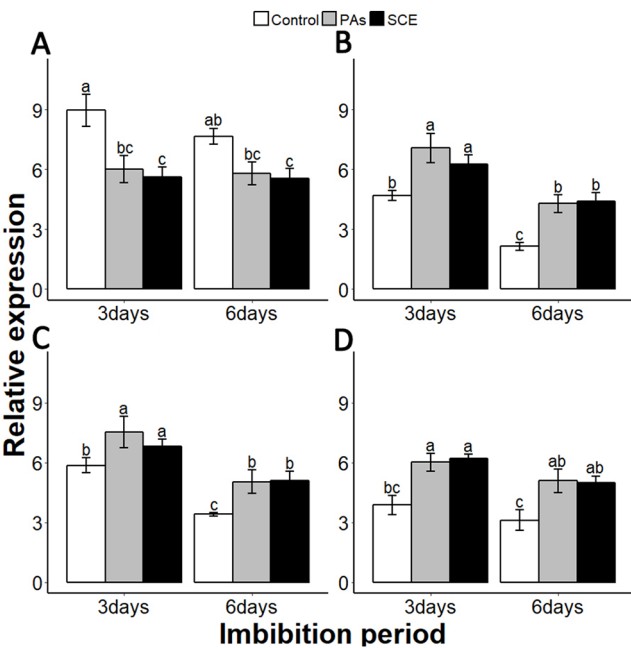

**Figure 6 Effect of SCE on the expression of GA-related genes.** (A), *SsGA3OX1* (B), *SsGAOX* (C), *SsRGL2* and (D), *SsGAI*. The expression of *SsGA3OX1, SsGAOX, SsRGL2* and *SsGAI* was determined by qRT-PCR on third and sixth days after treatment. *SsACTIN* was used as the reference gene. Control, seed grown in half MS medium. PAs, proanthocyanidins-supplemented (0.1%) half MS medium. SCE, half MS medium supplemented with seed-coat extract (0.3%). Data shown are means ± SD ($n = 3$). Means with different letters are significantly different at $P < 0.05$ using Tukey's HSD post hoc test.

treatments decreased the expression level of *SsCIPK23* gradually as compared to the control on both the third and sixth imbibition days (Fig. 7C).

We found interactions of seed coat PAs with the germination regulators. After seven days imbibition, the control, GA₃, NDGA, H₂O₂ and KNO₃ priming showed 36 ± 4.8, 97.22 ± 4.8, 91.9 ± 1, 93.44 ± 3 and 97.22 ± 4.8 (mean ± SD) percent germination respectively on 0.5× MS medium supplemented with 0.3% SCE. We found that all of our treatments significantly ($P = 0.05$) promoted germination as compared to the control (Fig. 8; Fig. S3).

## DISCUSSION

The data presented here show the causes of dormancy in *S. sebiferum* seed. PAs found in the seed coat and the ESC were the major controllers of dormancy in *S. sebiferum* seed. Our results show that *S. sebiferum* intact seeds are dormant, imbibe water and increase in germination if the covering structures such as the seed coat and endosperm cap are removed. Seeds showing this combination of features fit into the non-deep physiological dormancy category of *Baskin & Baskin (2014)*. The causes of dormancy in these seeds can lie in the covering structures like seed coat (*Baskin & Baskin, 2014*) and ESC. If the covering structures control dormancy, there are a number of known mechanisms by which they do so; they can be impermeable to water, they can mechanically constrain the embryo from germinating, they can contain chemical

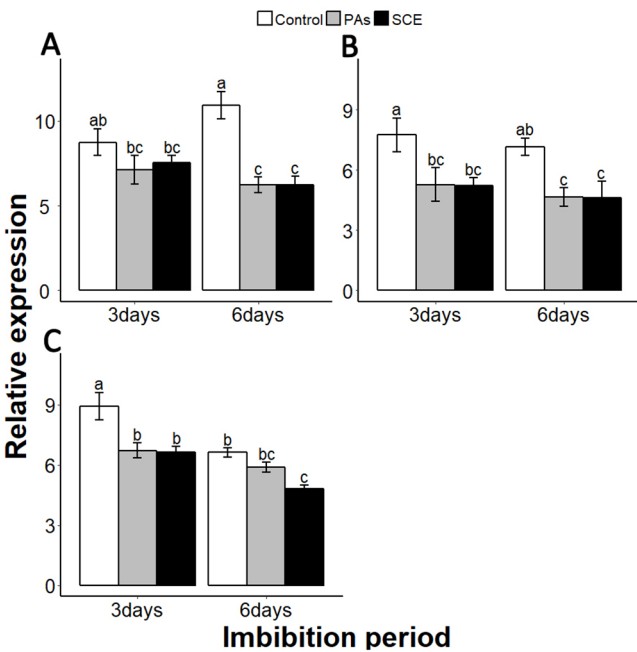

**Figure 7 Impact of SCE on expression levels of ROS- and nitrates-signalling genes.** (A), *SsMPK6* (B), *SsNLP8* and (C), *SsCIPK23*. The expression of *SsMPK6, SsNLP8* and *SsCIPK23* were determined by qRT-PCR on the third and sixth days after treatment. *SsACTIN* was used as the reference gene. Control, seeds grown in half MS medium. PAs, proanthocyanidins-supplemented half MS medium. SCE, seeds cultivated on half MS medium supplemented with seed-coat extract. Data shown are means ± SD ($n = 3$). Means with different letters are significantly different at $P < 0.05$ using Tukey's HSD post hoc test.

inhibitors of germination and they can prevent the exit of chemical inhibitors of germination present in the embryo.

*S. sebiferum* intact seeds take up water, and so the seed coat is permeable to water. These results are in agreement with the results of *Li et al. (2012)*. Recently, *McGill et al. (2017)* also reported that the rapid uptake of water within the first hour of imbibition indicates that the *Myosotidium hortensia* seed coat is not acting as a water impermeable barrier preventing germination, which suggests that water impermeability is not the reason for dormancy in *S. sebiferum* seed.

In our study, if the seed coat is removed, germination does increase substantially, as would be predicted from the model that the seed coat could have mechanical constraints. Our results show that mechanical scarification can slightly break the seed dormancy but scarification with sulphuric acid which degrades the germination inhibitors greatly improves germination. These results suggest that any mechanical constraint by the seed coat on germination is minor, compared to the major role played by germination inhibitors in the seed coat. Our results are in agreement with the results of *Wada, Kennedy & Reed (2011)*, which showed that the effectiveness of sulphuric acid for *Rubus* seed scarification was likely due to degradation of PAs in the testa. The acid both weakened the seed coat and degraded the inhibitors (PAs) which the seed coat contains, and these inhibitors restrict germination of decoated seeds. Our results suggest that covering

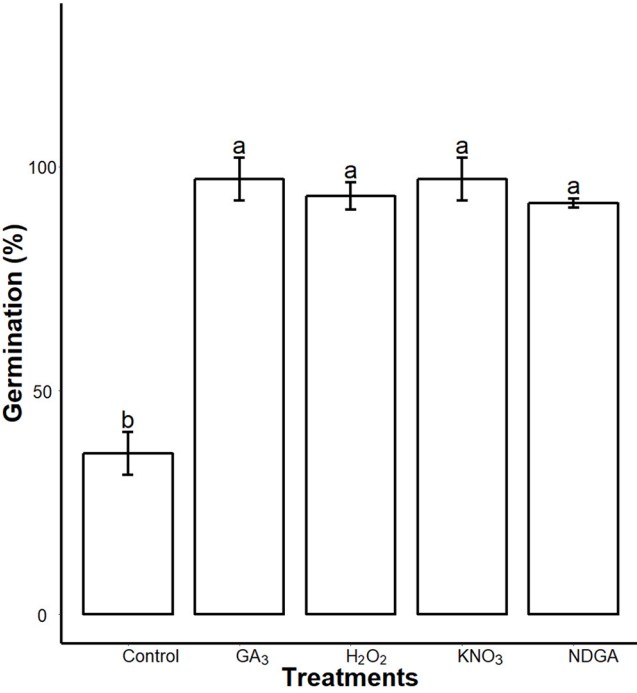

**Figure 8 Inhibitory effects of SCE on seed germination was alleviated by GA₃, NDGA, H₂O₂ and KNO₃.** Seeds were primed in double distilled water (Control), 50 μM GA₃, 50 μM NDGA, 20 mM H₂O₂ and 0.4% KNO₃ and were sowed in 0.3% SCE-supplemented half MS medium. The seed germination rate was recorded seven days after imbibition. Data shown are means ± SD ($n = 3$). Means with different letters are significantly different at $P < 0.05$ using Tukey's HSD post hoc test.

structures like tegmen contain germination inhibitors. These inhibitors restrict germination of decoated seeds, so removal of the seed coat results in 85% germination of the decoated seeds. The ESC also contains inhibitors, and the removal of the ESC results in 100% germination. It is also possible that the ESC is mechanically restraining the extra 15% of seeds from germinating, but the evidence we present here (vanillin assay) about the changes in the colour of the ESC support the chemical inhibition model.

Proanthocyanidins found in the tegmen and the ESC are the inhibitors of seed germination in *S. sebiferum* and have a molecular mechanism to inhibit seed germination. In our study, SCE (containing PAs) and pure PA treatments impacted the expression level of genes involved in GA/ABA homeostasis, which suggests that the seed-coat induced the seed dormancy by influencing transcription levels of ABA and GA biosynthesis or degradation genes and unbalance between GA/ABA homeostasis (*Debeaujon & Koornneef, 2000*; *Debeaujon, Leon-Kloosterziel & Koornneef, 2000*; *Liguo et al., 2012*). Seeds primed in GA₃ and NDGA recovered seed germination on a SCE-supplemented medium, which confirms the above-suggested model. ROS and nitrates are also important factors in seed germination due to their role in the maintenance of ABA/GA homeostasis (*Debeaujon & Koornneef, 2000*; *Jia et al., 2012*; *Lara et al., 2014*; *Liu et al., 2010*; *Yan et al., 2016*; *Zhou et al., 2015*). Exogenous application of PAs and SCE in growing medium significantly reduced the transcription level of ROS- and nitrates-signalling genes.

$H_2O_{2-}$ and $KNO_{3-}$ primed seed recovered the germination on SCE-containing medium. These results are in agreement with previously reported results that a mutation of Arabidopsis with transparent testa 8 (*TT8*) lacked PA accumulation in its testa, produced a high level of $H_2O_2$ after imbibition, and had higher germination rate (*Jia et al., 2013*; *Liu et al., 2010*). $H_2O_2$ is the main kind of ROS in plants which regulates seed germination through GA/ABA metabolism and signalling, (*Jia et al., 2012*, *2013*; *Liu et al., 2010*). KNO3 is a source of nitrates which activates the nitrate reductase enzyme and regulates the expression of nitrates-signalling genes *NLP8* and *CIPK23* which induce seed germination. *NLP8* binds directly to the promoter of *CYP707A2* and reduces ABA levels in a nitrate-dependent manner (*Footitt et al., 2017*; *Liu et al., 2010*; *Yan et al., 2016*).

## CONCLUSION

Our research offers a quick and easy method for germinating *S. sebiferum* seeds for nursery growers. Moreover, this experiment will provide a basis for researchers to understand the mechanisms involved in *S. sebiferum* seed dormancy. *S. sebiferum* seeds contain PAs in the seed coat (tegmen layer) and ESC. PAs impact the transcription of ABA-, GA-, ROS- and nitrates-related genes, and cause dormancy in *S. sebiferum* seed. In our experiments, we found dynamic changes in PA levels in the ESC. To generalize this result, more investigations of seeds of other species are required.

## ACKNOWLEDGEMENTS

The authors are thankful to Ghulam Ali Bugti, school of plant protection, Anhui Agriculture University, for his wise suggestions for improving the quality of manuscript.

### Funding

This work was funded by Anhui Natural Science Foundation (1708085QC70), the National Natural Science Foundation of China (11375232&31500531), the Science and Technology Service program of Chinese Academy of Sciences (KFJ-STS-ZDTP-002&KFJ-SW-STS-143-4), the Grant of the President Foundation of Hefei Institutes of Physical Science of Chinese Academy of Sciences (YZJJ201502&YZJJ201619), the major special project of Anhui Province (16030701103), and the research and technology project of Anhui province (1501031079). The funders had no role in study design, data collection and analysis, decision to publish, or preparation of the manuscript.

### Grant Disclosures

The following grant information was disclosed by the authors:
Anhui Natural Science Foundation: 1708085QC70.
National Natural Science Foundation of China: 11375232&31500531.
Science and Technology Service program of Chinese Academy of Sciences: KFJ-STS-ZDTP-002&KFJ-SW-STS-143-4.

Grant of the President Foundation of Hefei Institutes of Physical Science of Chinese Academy of Sciences: YZJJ201502&YZJJ201619.
Major special project of Anhui Province: 16030701103.
Research and technology project of Anhui province: 1501031079.

## Competing Interests

The authors declare that they have no competing interests.

## Author Contributions

- Faheem Afzal Shah conceived and designed the experiments, performed the experiments, contributed reagents/materials/analysis tools, prepared figures and/or tables, authored or reviewed drafts of the paper, approved the final draft, seed collection, Taking photographs.
- Jun Ni conceived and designed the experiments, contributed reagents/materials/analysis tools, prepared figures and/or tables, authored or reviewed drafts of the paper, approved the final draft, taking photographs.
- Jing Chen performed the experiments.
- Qiaojian Wang performed the experiments, contributed reagents/materials/analysis tools, seed collection.
- Wenbo Liu performed the experiments.
- Xue Chen analysed the data, contributed reagents/materials/analysis tools.
- Caiguo Tang analysed the data, contributed reagents/materials/analysis tools.
- Songling Fu conceived and designed the experiments, authored or reviewed drafts of the paper, approved the final draft.
- Lifang Wu conceived and designed the experiments, authored or reviewed drafts of the paper, approved the final draft.

## DNA Deposition

The following information was supplied regarding the deposition of DNA sequences:
    Gene sequences are provided in Data S1.

## Data Availability

    The raw data are provided in the Supplemental Dataset Files.

## Supplemental Information

Supplemental information for this article can be found online at http://dx.doi.org/10.7717/peerj.4690#supplemental-information.

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
