# Peer review of "Proanthocyanidins in seed coat tegmen and endospermic cap inhibit seed germination in Sapium sebiferum"

_PeerJ, doi:10.7717/peerj.4690_

## Round 0.1 · original submission · Major Revisions

Please, address the questions and comments of all 3 reviewers. Please, note also the suggestions in the attached review-pdf. Please, provide an improved version of the manuscript and an accompanying letter describing all changes in the manuscript.

·

Basic reporting

English is easy to understand but some grammatical errors are still spread all through the document and should be corrected.

The overall structure is correct, but the content within each section does not always match the sections (for instance, results stated in the introduction, methods detailed in the results section)

Experimental design

The design is not presented clearly and we understand parts of it, by mixing text of the material and method sections, text of the results section and elements in the figure and legends. A clear synthetic presentation is required.

No indications is provided regarding statistical analyses, although some statistical tests results are indicated on the figures.

Validity of the findings

It is difficult to judge as some elements are missing on the design description.

Additional comments

The manuscript needs thorough reworking to improve all sections of the documents, developing the state of the art, better presenting the objectives, completing the material and methods, centering the result section on the results by moving additional methodological details to the previous section, and focusing the discussion on the interpretation by synthesizing the statement of the conclusions rather than repeating detailed results.

Therefore, I recommend to reject the manuscript for publication, with possibility to submit again following completion of the modifications. This seems feasible, as some of the necessary elements can be found dispersed in the text at the wrong place or in the figure legends. It seems that no additional experimental work nor analyses should be done, but a huge effort to correctly and fully explain the design, the methods, the results.

·

Basic reporting

The standard of English writing in the paper is clear enough to be understandable, but needs editing and correcting in numerous places to improve clarity, especially for non-English readers. I have not given specific examples because there are so many.

The Introduction and background show the context and background, and give the aim(s) of the investigation. It would help the reader if some more detail of the internal structure of the seed was given, particularly how the endosperm is related to the embryo. The two layers of the seed coat (outer tegmen, inner testa) are described, but the position and structure of the endosperm relative to the embryo is not. From the photos supplied, it appears that the endosperm surrounds the embryo. If that is the case, the endosperm could potentially provide a mechanical constraint on the embryo, preventing germination, as is the case in other species. In this species, the results of the investigations suggest that this is not the case. But more detail about the endosperm is required here.

The structure basically conforms to PeerJ standards. There is a lot of repetition of material between sections, so that parts of results are given in the Introduction, parts of Methods are repeated in Results, and details of Results are repeated in Discussion. Examples of this are given in the Comments to Authors below.

Figures are clear and relevant. Figs 5 - 7 would benefit by the sub-figures being given letters (A – D), as in the preceding Figs, to assist referring to them in the text. That way, the long technical names for the genes (eg SsDOG1) can be left in the figures, and omitted from the text in Results. It would make the text in Results easier to read.

Experimental design

The authors have performed a carefully designed set of experiments to identify the cause of dormancy in Sapium sebiferum, and are to be commended for doing so.
They have framed a good research question which fills a gap in knowledge.
Methods are described in sufficient detail. The experiments have been well- conducted with replication and proceed to test explicit mechanisms of seed coat dormancy know from the literature.

Validity of the findings

Their experiments are analysed correctly and interpreted correctly. Their results support the hypothesis of germination inhibitors being present in the seed coat, and endospermic cap. There may be an element of mechanical constraint by the testa (see comments in Discussion). The Discussion as it stands is lengthy and would benefit from re-writing to remove repetition of details of results and present their case more clearly (see comments below).

Additional comments

Introduction:
Lines 57 – 58: the two layers of the seed coat are described (outer testa, inner tegment). Describe how the endosperm is related to the embryo (surrounds the embryo, lies alongside the embryo, or whatever is appropriate here)
Lines 63 – 73: this text is outline results, which should appear in Results. Delete from here.
Lines 61 – 62. Text about PA’s and nitrates: convert into a statement like ‘It is unclear whether PA’s respond to nitrate’

Methods
Line 104: ‘seeds were carefully uncoated with a scissor’. Explain that this procedure (better described as ‘decoated’ or ‘seed coat removal by dissection’) left the endosperm + embryo as a unit, hereafter referred to as ‘decoated seeds’, or ‘endosperm + embryo’. This makes it clear what the dissection of the seed coat left exposed. Where experiments were carried out with decoated seeds, make this clear in Methods.

Results:
Lines 139 – 143: this repeats the methods of the experiment, which were given in Methods. Do not repeat here. Start with ‘We found that …’
Line 150: while it is true that water uptake increased with scarification, it is very important to state there that the control seeds also took up water without any scarification (18% in Fig. 1B). This shows the seed coat is permeable to water.
Line 155: as the germination percentages are shown in Fig. 1C, there is no need to repeat them all in the text. It is worth stating here that germination of intact seeds was 2%, and ranged 40 – 65% in scarified seeds.
Line 164: again, gives the detail of the methods, and so is unnecessary repetition. Start with ‘We found that decoated seeds showed 85% germination …’
Lines 166 – 169: again, repetition of Methods. Start with ‘We found that seed coat extract (SCE) …’
Repetition of Methods is continued throughout Results. There is no need for it; the next important Result in this section is that SCE inhibited germination of decoated seeds.
Line 193 onwards gene expression section: this contains a lot of repetition of Methods, which needs deleting/moving to Methods. In reporting the results, the names of the genes are very lengthy and confusing. It would be better to report results such as the one in lines 199 – 202 as ‘The expression of a gene that positively regulates germination (SsDOG1) was significantly higher in SCE and PA treatments than in the control (Fig. 5)’. Describe the genes in terms of what they regulate, and put the long names in brackets. Alternatively, give the subfigures in Fig. 5 etc the letters A, B, C etc, and leave the long name on the Figures only (so refer to ‘Fig. 5A’ in the example above.
Line 194: ‘we found the crosstalk effect of seed coat with dormancy and GA, …’. Re-phrase as ‘We examined the relationship between seed coat dormancy and the gene expression of GA, ….’. ‘crosstalk effect’ is used also in line 239, and this should be replaced by ‘relationship between’ or similar expression.


Discussion

The Discussion is fairly long as it stands, and would be easier to follow if reduced. It could be shortened by using the framework suggested below. This sets up a logical framework based on known mechanisms of seed coat dormancy as set out in Baskin and Baskin (2014), that the results from S. sebiferum can be easily related to.


The Results for S. sebiferum show intact seeds are dormant, imbibe water, and increase in germination if the embryo is removed from covering structures such as the seed coat and/or endosperm. Seeds showing this combination of features fit into the non-deep physiological dormancy category of Baskin and Baskin (2014). The causes of dormancy in these seeds can lie in the covering structures, the embryo, or in a combination of both (Baskin and Baskin 2014).

If the covering structures control dormancy, there are a number of known mechanisms by which they do so: the covering structures are:
- impermeable to water
- mechanically constrain the embryo from germinating
- contain chemical inhibitors of germination
- prevent the exit of chemical inhibitors of germination present in the embryo
- restrict oxygen diffusion to the embryo


The results the authors have can eliminate some of these mechanisms:

Water permeability: the intact seeds take up water and so the seed coat is permeable to water; if the seed coat was impermeable, water uptake of intact seeds in Fig. 1B would be effectively zero.

Mechanical constraint: if the seed coat is removed, germination does increase substantially, as would be predicted from this model. Scarification of the seed coat with sulphuric acid improved germination. The acid both weakened the seed coat, and degraded the PCA’s which the seed coat contains, and these inhibitors restrict germination of decoated seeds (see below). So the seed coat, particularly the testa, may be exerting mechanical constraint on germination. The authors cannot eliminate this possibility with results they have, but, this could be tested by trying a non-chemical form of scarification (if possible), to see whether physically disrupting the testa resulted in improved germination. If it did, the seed coat would appear to be exerting mechanical constraint. If physical scarification does not improve germination, then the germination inhibitors in the seed coat model are restricting germination.

Covering structures contain germination inhibitors: the authors have built a strong case that there are germination inhibitors in the seed coat, that these inhibitors restrict germination of decoated seeds, that the endospermic cap also contains inhibitors, and that while removal of the seed coat results in 85% germination of the decoated seeds, removal of the endospermic cap results in 100% germination (ie, all the remaining seeds). Moreover, the restricting effects of the seed coat extract and PCA can be reversed by application of GA3, H2O2 etc to decoated seeds. And the gene expression data support the chemical inhibitor model. It is also possible that the endospermic cap is mechanically restraining the extra 15% of seeds from germinating, but the evidence presented of changes in the colour (PCA content) of the endospermic cap support the chemical inhibition model.

Oxygen diffusion model: this is hard to test. But the evidence above for the chemical inhibitor model makes limitation of oxygen diffusion by the covering structures unlikely

Prevention of exit of inhibitors from the embryo: again, unlikely given the evidence for the presence of inhibitors in the seed coat and endospermic cap.



If the above structure is used for Discussion, it will make it more coherent and easier for the reader to see how the evidence from the investigation matches against known dormancy mechanisms.

Reviewer 3 ·

Basic reporting

Seed dormancy in plants is one of the bright examples of contradictions between natural selection and adaptation of plants to unfavorable environmental conditions on the one hand and artificial selection and breeding purposes tasks on the other hand. One of the ways to manipulate with seed dormancy is based on knowledge of genes underlying this trait. The results of biochemical, physiological and genetic investigations of seed dormancy in Chinese tallow, described in the considered article, are of importance.

Experimental design

The article represents original primary research within aims and scope of the journal. Methods described with sufficient detail & information to replicate. I suggest to add description of methods of statistical analysis (in the current version these methods are mentioned in figures legends only).

Validity of the findings

Data is robust, statistically sound, & controlled.

Conclusions are well stated,

Additional comments

Seed dormancy in plants is one of the bright examples of contradictions between natural selection and adaptation of plants to unfavorable environmental conditions on the one hand and artificial selection and breeding purposes tasks on the other hand. One of the ways to manipulate with seed dormancy is based on knowledge of genes underlying this trait. The results of biochemical, physiological and genetic investigations of seed dormancy in Chinese tallow, described in the considered article, are of importance.
The article can be published after some revision.

1. Article title: I recommend to change to “Proanthocyanidins in seed coat’s tegmen and endospermic cap inhibit seed germination in Sapium sebiferum”.
Chinese tallow is first of all the ornamental plant and a source of oil used for some industrial purposes. Yes, it is considered as a promising energy plant, but it is not sufficient to call it “the bioenergy plant” in the article title. It is the overstatement.

2. Introduction.
Lines 40-42. Not some, but many plants contain proanthocyanidins in seed coats, and the relationship between PAs in seed coats and seed dormancy was first discovered in wheat. In the model plant Arabidopsis, mentioned in lines 40-42, this relationship was described much later. I think it is important to say in the beginning of the Introduction section that the role of PAs for seed dormancy was first discovered in wheat. As early as in 1914, Nilsson-Ehle showed that red seed coat color in wheat is associated with extended dormancy period in comparison to that in white-grained wheat. In 1958, Miyamoto&Everson showed that red pigment of seed coat is a substance derived from catechins (PAs).

3. Materials and methods:
Line 135. Here methods of statistical analysis should be described.

4. Conclusions
Some practical conclusion are required. How the knowledge obtained in the current study can be used in breeding.

5. Supplementary materials
I did not find description of supplementary materials, supplementary figures legends, headings of tables presented in excel format, etc.

6. English
The English language should be improved (over the whole text). Some examples where the language could be improved include Line 18 (“highly” should be removed or replaced), Lines 96-98 (Seed coat proanthocyanidins contents were analyzed <here preposition is needed> conventional HCl–vanillin assay (Herald et al. 2014) and pure proanthocyanidins (UV≥95%) was <”was” should be replaced by “were”> used as a reference.), Line 93 (“was” should be replaced by “were”), Lines 372-374 – the current phrasing (and using “…which may …” 2 times in 1 sentence) makes comprehension difficult.

7. Additional changes:
S. Sebiferum should be changed to S. sebiferum (over the whole text).

---

## Round 0.2 · Minor Revisions

The revised manuscript has greatly improved and is close to acceptance. One reviewer has some suggestions for further improvement of the manuscript. Please, add those suggestions in a revised manuscript.

·

Basic reporting

Text is greatly improved; specific comments (most minor) are given below. Repetition of text present in the earlier version has largely been eliminated. The structure of the Introduction and Discussion is improved greatly.

Experimental design

Sound: as per my previous review.

Validity of the findings

The experiments are analysed correctly and interpreted correctly. Their results support the hypothesis of germination inhibitors being present in the seed coat, and endospermic cap. There may be an element of mechanical constraint by the testa.

Additional comments

Specific comments to authors

Lines 37 – 40: this currently reads as though the hard (water impermeable) seed coat plays a role in all of the other seed coat dormancy mechanisms. It would be clearer to re-word these lines as eg. ‘The seed coat can play a role in regulating dormancy. Know mechanisms by which the seed coat regulates dormancy include the prevention of water uptake by an impermeable seed coat, or inhibiting gas exchange (McGill citation), or by mechanically constraining the embryo, or by containing germination inhibitors (B & B citation’.

Line 48: italicise Rubus

Line 52: PA’s located in the seed coat can …’ (replace ‘distribution’ with ‘located’)

Line 72: insert ‘outer’ before ‘testa’ and insert ‘inner’ before ‘tegmen’. This reminds readers of the relative positions of the two seed coat layers.

Line 73: Better re-worded as ‘The tegmen encloses the endosperm, which in turn encloses the embryo’.

Line 73. ‘Tallow tree seeds had good water permeability, but the seed coat structure at the side of the radicle …’ replace with ‘Tallow tree seeds readily imbibed water but the seed coat at the site of the radicle appeared to ….’

Line 79: insert ‘germination’ in front of ‘inhibitors’

Line 79: Re-word ‘Which layer of the seed coat …’ as ‘It is currently unknown which layer of the seed coat …’

Line 83: re-word ‘Can the application of an exogenous ABA …’ as ‘We tested whether exogenous application of an ABA …’

Line 87: replace ‘unveil’ with ‘demonstrate’

Line 103: replace ‘was purchased’ with ‘were purchased’

Lines 121 – 2: delete sentence ‘All things were weighed by …’ (unnecessary detail, can be omitted)

Line 145: replace ‘scar’ with ‘cut’

Line 148: replace ‘replicons’ with ‘replicates’

Lines 191 – 5: the text ‘S. seberiferum has a hard seed coat … and seedling growth’ is all repetition of information given previously in the Introduction and Methods, and can be deleted. The heading for this section indicates what experiment is being talked about. Start text with ‘We found that sulphuric acid …’

Line 201 - 2: it is very important here to state that the untreated seeds themselves showed substantial water uptake, and the current text does not draw this important finding out. Re-word as eg ‘Water uptake of untreated control seeds was 16 – 17% (authors to give the %) over X hours; seeds treated with sulphuric acid for longer periods of time showed increasing water uptake to a maximum of 20% (authors check %) after 60 minutes of acid treatment…’ Then give details of significant differences as currently done.

Line 206: insert ‘from’ after ‘ranged’ (ranged from 40 – 60% …)

Line 222: insert ‘a’ in front of ‘dark brownish colour …’

Line 234: change ‘We examine …’ to ‘We examined …’

Line 292: replace ‘played the main role in ‘ with ‘were the major controllers of dormancy in …’

Line 296: the listing of possible mechanisms of seed coat dormancy need to have ‘can’ inserted in front of each one: ‘..they can be impermeable to water, they can mechanically constrain … can contain chemical inhibitors ….can prevent the exit of …’

Line 305: replace ‘this model’ with ‘the model’

Line 307: insert ‘which degrades the germination inhibitors ..’ after ‘scarification with sulphuric acid’, to read ‘but scarification with sulphuric acid which degrades the germination inhibitors greatly improves germination’.

Lines 307- 8: the conclusion here about the relative strengths of mechanical constraint vs germination inhibitors could be re-worded as ‘These results suggest that any mechanical constraint by the seed coat on germination is minor, compared to the major role played by germination inhibitors in the seed coat’.

Reviewer 3 ·

Basic reporting

Text is thoroughly revised

Experimental design

The statistical methods are described now

Validity of the findings

Data is robust, statistically sound, & controlled.

Additional comments

The authors have made thoroughly the necessary changes. Article in its current form is acceptable for publiction.

---

## Round 0.3 · accepted · Accept

Congratulations on the acceptance of your manuscript.

#